# TableRAG: Million-Token Table Understanding with Language Models

**Si-An Chen**[1]*, **Lesly Miculicich**[2], **Julian Martin Eisenschlos**[3],
**Zifeng Wang**[2], **Zilong Wang**[4]*, **Yanfei Chen**[2], **Yasuhisa Fujii**[3],
**Hsuan-Tien Lin**[1], **Chen-Yu Lee**[2], **Tomas Pfister**[2]

[1]National Taiwan University, [2]Google Cloud AI Research,
[3]Google DeepMind, [4]UC San Diego

## Abstract

Recent advancements in language models (LMs) have notably enhanced their ability to reason with tabular data, primarily through program-aided mechanisms that manipulate and analyze tables. However, these methods often require the entire table as input, leading to scalability challenges due to the positional bias or context length constraints. In response to these challenges, we introduce TableRAG, a Retrieval-Augmented Generation (RAG) framework specifically designed for LM-based table understanding. TableRAG leverages query expansion combined with schema and cell retrieval to pinpoint crucial information *before* providing it to the LMs. This enables more efficient data encoding and precise retrieval, significantly reducing prompt lengths and mitigating information loss. We have developed two new million-token benchmarks from the Arcade and BIRD-SQL datasets to thoroughly evaluate TableRAG's effectiveness at scale. Our results demonstrate that TableRAG's retrieval design achieves the highest retrieval quality, leading to the new state-of-the-art performance on large-scale table understanding. The implementation and dataset will be available at https://github.com/google-research/google-research/tree/master/table_rag.

## 1 Introduction

Recent advancements have leveraged language models (LMs) for table understanding tasks [5]. This typically involves prompting LMs with entire tables to ensure thorough analysis [24, 3, 11, 22, 10, 29]. However, scaling to larger tables poses several challenges. First, LMs face context-length constraints; for example, a medium-sized table with 100 columns and 200 rows translates into over 40,000 tokens, surpassing the limits of popular LMs like LLaMA and the GPT series. Additionally, long contexts can degrade reasoning capabilities, a phenomenon known as the *Lost-in-the-Middle* [9]. Finally, computation costs and latency increase significantly with table size. Therefore, developing a scalable solution that efficiently handles large tables remains a critical area of research.

Naive approaches to making large table understanding feasible, such as truncating the table or only reading the schema, often result in the loss of critical information. To address this, previous works have attempted to retrieve key rows and columns to construct a sub-table that captures essential information for answering queries [8, 17]. These methods encode entire rows and columns into sparse or dense embeddings to reduce token costs for LMs, yet they still face computational and performance challenges with extremely large tables. Encoding entire rows and columns requires processing the whole table, which is infeasible for large tables containing millions of cells. Furthermore, compressing

---

*Work done while the author was a student researcher at Google Cloud AI Research. Correspondence to: Si-An Chen<sianchen.kevin@gmail.com>, Chen-Yu Lee<chenyulee@google.com>

38th Conference on Neural Information Processing Systems (NeurIPS 2024).

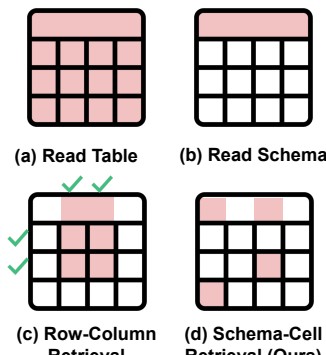
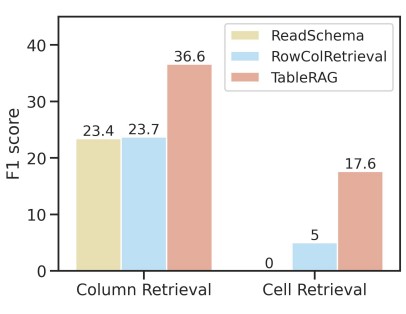

Figure 1: Comparison between table prompting techniques for LMs. (a) - (d): Data included in the LM prompt (shaded region). (a) **Read Table**: The LM reads the entire table, which is often infeasible for large tables. (b) **Read Schema**: The LM reads only the schema, consisting of column names and data types, resulting in a loss of information from the table content. (c) **Row-Column Retrieval**: Rows and columns are encoded and then selected based on their similarity to the question. Only the intersection of these rows and columns is presented to the LM. It is still infeasible to encode all rows and columns for large tables. (d) **Schema-Cell Retrieval** (our work): Column names and cells are encoded and retrieved based on their relevance to LM-generated queries about the question. Only the retrieved schema and cells are provided to the LM, enhancing efficiency in both encoding and reasoning. (e) Retrieval results on the ArcadeQA dataset show that TableRAG outperforms other methods in both column and cell retrieval, thereby enhancing the subsequent table reasoning process. The **Read Table** technique is excluded as reading entire tables is typically infeasible in this context.

long rows and columns into a fixed-size embedding can obscure semantic meaning, particularly when the table contains diverse or less semantically rich content (e.g., numerical values).

In this work, we introduce, TableRAG, a scalable framework that leverages retrieval-augmented generation (RAG) for LM-based table understanding. We illustrate the key differences between prior table prompting approaches and the proposed TableRAG in Fig. 1.

Our method integrates schema retrieval and cell retrieval to extract essential information from tables, enabling a program-aided LM agent to solve queries based on the provided information. Schema retrieval allows LMs to identify crucial columns and their data types solely by column names, avoiding the need to encode entire columns. Cell retrieval enables the identification of keywords for indexing or pinpointing columns that contain necessary but hard-to-find information missed by schema retrieval alone. To build the database for cell retrieval, TableRAG encoded each cell independently, addressing the issue faced when encoding entire rows and columns. Furthermore, TableRAG only encodes distinct and the most frequent categorical values, reducing the encoder's token cost (as shown in Fig. 3) and operating within a user-specified budget effectively. Both retrieval processes in TableRAG are enhanced by query expansion [21] with dedicated prompts for schema retrieval and cell retrieval, ensuring thorough and relevant data extraction.

Existing TableQA benchmarks typically feature only small tables with dozens of rows and columns. To verify the scalability of TableRAG for larger tables, we build two new benchmarks sourced from the real-world Arcade [26] and BIRD-SQL [7] datasets. Additionally, to assess performance across various scales, we generated synthetic data expanding tables from the TabFact dataset to larger sizes, while maintaining consistent questions and key table content for evaluation. Our experimental results demonstrate that TableRAG outperforms existing table prompting methods significantly and consumes fewer tokens across different table sizes.

Our contributions can be summarized as follows:

- We conduct the first extensive study exploring the application of LMs to large-scale, real-world tables, analyzing the scalability and limitations of existing LM-based table reasoning approaches.

- We introduce two new real-world benchmarks derived from Arcade and BIRD-SQL, along with an expanded synthetic dataset from TabFact. These datasets include tables ranging from tens to

millions of cells (Table 7), enabling comprehensive evaluation of LM capabilities across various table scales.

- We develop TableRAG, an efficient framework for LM-based table understanding that demonstrates superior performance on large tables while minimizing token consumption. We conduct a detailed ablation study to validate the effectiveness of each component within TableRAG.

## 2 Related Work

Research in table understanding has evolved from fine-tuning specialized architectures [6, 4] to leveraging LMs in few-shot setups [22, 10], capitalizing on the emerging reasoning capability of these models. To enable LMs to understanding tables, table information must be included in the prompts. Representative works like Dater [24], Binder [3] and subsequent works [22, 29, 11, 10, 28] typically require LMs to process entire tables. These methods, while effective in leveraging LMs' reasoning and programming capabilities for question answering, are often not feasible for larger tables due to the context length constraints.

To address the limitations of processing full tables, two main streams have emerged: schema-based and row-column retrieval methods. Schema-based methods, such as Text2SQL [30] and more recent developments [20, 18, 13, 14], focus primarily on schema understanding to generate SQL commands. This significantly reduces token complexity but at the cost of omitting valuable cell data. Row-column retrieval methods, such as ITR [8] and TAP4LM [17], attempt to address scalability issues by encoding and retrieving essential rows and columns. While this strategy reduces input lengths for reasoning, it still requires substantial computation to encode entire rows and columns and can suffer from poorer embedding quality for long sequences.

Our approach, TableRAG, innovates by combining schema retrieval with selective cell value retrieval and frequency-aware truncation. This creates an efficient table prompting method where the input length to LMs is independent of table sizes. This strategy significantly reduces computational demands while preserving the benefits of accessing table contents, leading to superior performance compared to other methods across various scales.

## 3 TableRAG

### 3.1 Motivation

An illustration of the workflow of our method is shown in Fig. 2. The core idea is to incorporate schema retrieval and cell retrieval to obtain necessary information for solving the questions by program-aided LMs. In practice, we often find that processing the entire table for LMs is unnecessary. Instead, the critical information usually lies in specific column names, data types, and cell values that directly relate to the question. For example, consider the question *"What is the average price for wallets?"* To address this, a program may simply need to extract rows related to "wallets" and then calculate the average from the price column. Knowing just the relevant column names and how "wallets" are represented in the table suffices to write the program. Our method, TableRAG, leverages the observation and addresses the context length limitations by RAG.

### 3.2 Problem Formulation

In large-scale table understanding, we are presented with a table $T$, represented as $T = v_{ij} \mid i \leq N, j \leq M$, where $N$ is the number of rows, $M$ is the number of columns, and $v_{ij}$ is the cell value at row $i$ and column $j$. We address a natural language question $Q$ and aim to produce an answer $A$ by an LM $\mathcal{L}$. Given the often impractical size of $T$ for direct processing, a table prompting method $P$ is employed to transform $T$ into a more manageable prompt $P(T)$, allowing $\mathcal{L}$ to effectively generate the answer $A = \mathcal{L}(P(T))$. Our objective is to develop an efficient $P$ that significantly reduces the size of the prompt, $|P(T)|$, compared to the original table, $|T|$, making it feasible for the LM to process large tables.

### 3.3 Core Components of TableRAG

**Tabular Query Expansion** To effectively manipulate the table, it is essential to pinpoint the precise column names and cell values necessary for the query. Unlike previous works [8, 17] that may use the question as a single query, we propose generating separate queries for both schema and cell values. For instance, for a question like *"What is the average price for wallets?"*, the LM is prompted

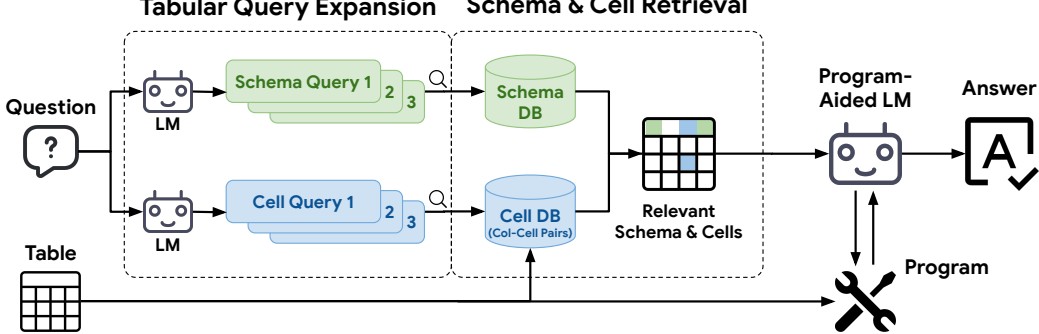

Figure 2: Workflow of the TableRAG Framework. The table is utilized to build the schema and cell databases. A question is then expanded into multiple schema and cell queries by LMs. These queries are sequentially utilized to retrieve schemas and column-cell pairs. The top $K$ candidates from each query are combined and fed into the LM solver's prompt to answer the question. The pseudocode and an answering example on ArcadeQA can be found in Alg. 1 and Fig. 8 respectively.

to produce potential queries for column names such as "product" and "price", and for relevant cell values like "wallet". These are then used to retrieve relevant schema and cell values from the table.

**Schema Retrieval** Following the generation of queries, the schema retrieval fetches pertinent column names using a pre-trained encoder $f_{\text{enc}}$, which encodes the queries and matches them against the encoded column names to determine relevance. The retrieved schema data includes column names, data types, and example values. We convert columns to integer, float, or datetime data types when feasible; otherwise, we keep them as categorical columns. For columns identified as numerical or datetime data type, we display the minimum and maximum values as example values. Conversely, for categorical columns, we present the three most frequent categories as example values. We combine the top-$K$ retrieval results for each query and rank them by their similarity to the closest query. The retrieved schema provides a structured overview of the table's format and content that will be used for more targeted data extraction.

**Cell Retrieval** Following schema retrieval, we proceed to extract specific cell values needed to answer the question. This involves building a database of distinct column-value pairs from $T$, denoted as $V = \bigcup_{ij}(C_j, v_{ij})$, where $C_j$ is the column name of the $j$-th column. In practice, the set of distinct values is often much smaller than the total number of cells, as illustrated in Fig. 3. This discrepancy significantly enhances the efficiency of cell retrieval.

Cell retrieval plays an crucial role in TableRAG. It enhances LM's table understanding capabilities in:

1. Cell Identification: It allows LMs to accurately detect the presence of specific keywords within the table, which is essential for effective indexing. For example, it can distinguish between terms like "tv" and "television", ensuring that searches and operations are based on precise data entries.

2. Cell-Column Association: It also enables LMs to associate particular cells with their relevant column names. This is crucial when questions pertain to specific attributes, such as linking the term "wallet" directly to the "description" column, thereby enables row-indexing.

It should be noted that cell retrieval is primarily beneficial when indexing by cell values is required. In other scenarios, simply knowing the schema may suffice. For example, to answer the question "What is the average price?", identifying the relevant column name for prices is sufficient because the actual computation of the average can be handled programmatically. Nevertheless, cell retrieval still improves TableRAG with additional key values from the table, which will be shown in Sec 4.8.

**Cell Retrieval with Encoding Budget** In the worst case, the number of distinct values could match the total number of cells. To maintain the feasibility of TableRAG in such cases, we introduce a cell encoding budget $B$. If the number of distinct values exceeds $B$, we restrict our encoding to the $B$ most frequently occurring pairs, thus improving efficiency when processing large tables. It is important to note that the encoding budget impacts only the cell retrieval process. Even if a cell is not included for retrieval, the subsequent solver can still access the cell if its column name is known

Table 1: Token complexities of primary table prompting approaches without truncation. Note that Read Schema does not aware of any cell content. $N$: number of rows, $M$: number of columns, $K$: number of top retrieval results, $D$: number of distinct values in the table. It is generally observed that $K < M \ll D \ll NM$.

| Table Prompting Approach | Methods | Token Complexity |
|---|---|---|
| Read Table | Dater [24]
Binder [3]
MultiTableQA [11]
Chain-of-Table [22]
Mix-SC [10]
TableLlama [28]
ReAcTable [29] | Reasoning: $O(NM)$ |
| Read Schema | MAC-SQL [20]
SQL-PaLM [18]
DIN-SQL [13]
DTS-SQL [14] | Reasoning: $O(M)$ |
| Row-Col Retrieval | ITR [8]
TAP4LM [17] | Encoding: $O(NM)$
Reasoning: $O(K^2)$ |
| Schema-Cell Retrieval | TableRAG (our work) | Encoding: $O(D)$
Reasoning: $O(K)$ |

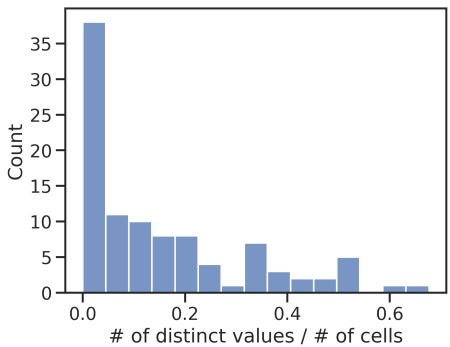

Figure 3: Histogram of the proportion of number of distinct values to number of cells in ArcadeQA and BirdQA. The figure indicates that for most tables, the number of distinct values ($D$) are much smaller than the number of cells ($NM$).

through schema retrieval or other cells. For instance, as shown in Fig. 8, the "description" column contains free-form text, which likely results in a high number of unique values, many of which may be truncated due to the cell encoding budget. Nevertheless, as long as the solver recognizes the column, it can still perform operations on that column to extract the required information.

**Program-Aided Solver** After obtaining the column names and cell values relevant to the question, the LMs can use these information to effectively interact with the table. TableRAG is compatible with LM agents which can interact with tables programmatically. In this work, we consider ReAct [23], which is a popular approach to extend the capability of LMs and has been used in recent literature to achieve state-of-the-art results in Table QA benchmarks [10, 29]. An example of how TableRAG works with ReAct is illustrated in Fig. 8.

### 3.4 Token Complexity Analysis

The efficiency and latency of invoking LMs are significantly influenced by the number of input tokens. Therefore, we focus on analyzing *token complexity*, which refers to the complexity introduced by the number of tokens in the LM's input. We examine the token complexity of each operation within the TableRAG framework and other table prompting methods. We assume that the length of a column name, a cell value, and the question are all $O(1)$. Note that $N$ represents the number of rows, $M$ represents the number of columns, $D$ is the number of distinct text cell values, $B$ is the cell encoding budget, and $K$ is the number of top retrieval results.

The token complexity of primary table prompting approaches are shown in Table 1.

- **Read Table** feeds the entire table to LMs, resulting $O(NM)$ tokens for reasoning.
- **Read Schema** only feeds the schema to LMs, which is $O(M)$ tokens, but loss the information from the table content.
- **Row-Column Retrieval** encodes all rows and columns to embeddings, resulting $O(NM)$ tokens for encoding. Then it retrieves top-$K$ rows and columns to construct a $K \times K$ sub-table for reasoning, which is $O(K^2)$.

We analyze the token complexity of each step in TableRAG as follows:

- **Tabular Query Expansion**: The prompt to the LM is primarily based on the question, which typically comprises only a few words, thus the token complexity for this component is $O(1)$.
- **Building Schema Database**: Each column name is encoded using the encoder function $f_{\text{enc}}$, resulting in a token complexity of $O(M)$ for the encoder.
- **Building Cell Database**: This operation involves encoding distinct column-value pairs using $f_{\text{enc}}$. The total number of distinct pairs $D$ is capped at $B$ when exceeding the limit. Therefore, the token

complexity for building the cell database is $O(\min(D, B))$, ensuring that the most frequent data is processed to optimize performance. Note that it takes $O(NM)$ CPU time to build the cell database, but the computation cost is negligible compared to LM calls. For example, computing the distinct values from $10^6$ elements takes only $60$ ms on a personal laptop, while it takes $1$ second to encode merely hundreds of words on CPUs[2].

- **LM Reasoning**: The query expansion process generally produces approximately 3-5 queries, which are considered $O(1)$. Each query then retrieves the top-$K$ results, leading to a total complexity of $O(K)$ for the columns and cell values included in the LM's prompt. This step ensures that the LM processes only the most pertinent information from the table, enhancing efficiency and effectiveness in generating responses.

Overall, given that typically $M \ll B$ and $D$, the token complexity for TableRAG is $O(\min(D, B))$ for encoding and $O(K)$ for reasoning, with neither being dependent on the overall size of the table. Consequently, TableRAG maintains manageable costs even for large tables, optimizing computational resources and response times for large-scale table understanding tasks.

## 4 Empirical Studies

### 4.1 Dataset

Existing TableQA benchmarks such as TabFact [2] and WikiTableQA [12] feature small web tables that fall within the context limits of most LMs, while others like Spider dataset [27] are designed purely as text-to-SQL tasks with access only to the schema. To better assess reasoning capabilities over larger, more realistic tables, we have developed two extensive table QA datasets, ArcadeQA and BirdQA, derived from the Arcade [26] and BIRD-SQL [7] datasets, respectively. ArcadeQA comprises tables with an average of $79,000$ rows and a maximum of 7 million cells, while BirdQA tables feature an average of $62,000$ rows with a peak at 10 million cells. Furthermore, we expanded TabFact to include synthetic tables ranging from $100 \times 100$ to $1,000 \times 1,000$, equivalent to a million of cells, to examine the impact of different table sizes under the same question and key information. Detailed methodology for dataset generation and the statistics of these datasets are provided in Appendix C and summarized in Table 7.

### 4.2 Baseline

We compare TableRAG to four different approaches for incorporating table information into LMs. To ensure a fair comparison of table access strategies, we have implemented the baselines and our TableRAG based on the same PyReAct [10, 29] solver.

**ReadTable** This common approach in recent research includes embedding the entire table in the prompt, then asking the LMs to solve the problem with the provided table information. This approach is limited by the context length of the LMs. We discard data instances and consider them failures when the table size exceeds the context length.

**ReadSchema** Widely-used in Text2SQL literature, this method assumes that table content is not directly accessible. It incorporates column names and data types into the prompt, enabling LMs to execute commands based on these column names without direct access to the row data.

**RandRowSampling** This method is a prevalent rule-based sampling approach [17] that randomly selects rows from the table with equal probabilities. When the total number of rows exceeds $K$, we select $K$ rows to form a representative sample. This baseline is employed to underscore the benefits of more targeted retrieval methods, illustrating how they can provide more relevant and efficient data selection compared to random sampling.

**RowColRetrieval** This state-of-the-art approach reduces table sizes prior to LM reasoning. Following the methodology of Sui et al. [17], we encode rows and columns and then retrieve the top $K$ rows and columns based on their similarity to the question's embedding to form a sub-table. Since encoding all rows and columns requires $2NM$ tokens, which is impractical for large tables with millions of cells, we truncate the tables to $\frac{B}{2M}$ rows. This truncation limits the number of tokens encoded to $B$, aligning with the token limit in our TableRAG implementation.

---

[2]https://kennethenevoldsen.github.io/scandinavian-embedding-benchmark/

Table 2: Performance comparison of table prompting approaches on ArcadeQA and BirdQA across LMs.

| Method | GPT 3.5 Turbo | | Gemini 1.0 Pro | | Mistral Nemo | |
| | ArcadeQA | BirdQA | ArcadeQA | BirdQA | ArcadeQA | BirdQA |
|---|---|---|---|---|---|---|
| ReadTable | 4.6 | 9.1 | 1.5 | 4.9 | 5.4 | 8.4 |
| ReadSchema | 43.1 | 40.3 | 30.8 | 31.8 | 32.3 | 35.7 |
| RandRowSampling | 42.3 | 34.7 | 20.8 | 25.3 | 28.5 | 33.4 |
| RowColRetrieval | 37.7 | 39.6 | 16.9 | 21.8 | 30.0 | 36.4 |
| **TableRAG** | **49.2** | **45.5** | **42.3** | **44.2** | **46.2** | **45.1** |

Table 3: Evaluation of retrieval performance. TableRAG shows best retrieval quality on all tasks. R: recall, P: precision.

| Method | ArcadeQA | | | | | | BirdQA | | | | | |
| | Column Retrieval | | | Cell Retrieval | | | Column Retrieval | | | Cell Retrieval | | |
| | R | P | F1 | R | P | F1 | R | P | F1 | R | P | F1 |
|---|---|---|---|---|---|---|---|---|---|---|---|---|
| ReadSchema | **100.0** | 12.4 | 23.4 | 0.0 | 0.0 | 0.0 | **100.0** | 30.8 | 41.6 | 0.0 | 0.0 | 0.0 |
| RandRowSampling | **100.0** | 12.4 | 23.4 | 66.5 | 0.5 | 3.5 | **100.0** | 30.8 | 41.6 | 48.9 | 1.7 | 7.2 |
| RowColRetrieval | 99.6 | 12.5 | 23.7 | 62.8 | 0.6 | 5.0 | 93.5 | 31.1 | 42.4 | 52.7 | 4.6 | 14.0 |
| **TableRAG** | 98.3 | **21.2** | **36.6** | **85.4** | **3.4** | **17.6** | 95.3 | **36.0** | **48.8** | **87.4** | **5.7** | **17.3** |

## 4.3 Experimental Setup

Our experiments employ GPT-3.5-turbo [1], Gemini-1.0-Pro [19] and Mistral-Nemo-Instruct-2407[3] as LM solvers. In ablation study, we use GPT-3.5-turbo if not specified. We use OpenAI's text-embedding-3-large[4] as the encoder for dense retrieval. For TableRAG, we set the cell encoding budget $B = 10,000$ and the retrieval limit $K = 5$. For RandRowSampling and RowColRetrieval, we increase the retrieval limit to $K = 30$. Each experiment is conducted 10 times and evaluated by majority-voting to ensure the stability and consistency. The evaluation metric is the exact-match accuracy if not specified.

## 4.4 Main Results

In evaluations across the datasets shown in Table 2, TableRAG consistently outperformed other methods, achieving the highest accuracies across all LMs on both ArcadeQA and BirdQA. The ReadTable method underperforms on both in ArcadeQA and BirdQA, indicating it suffers from long context. Among the three LMs, GPT 3.5 Turbo consistently delivers the best performance, regardless of the table prompting method used. These results demonstrate the effectiveness of TableRAG in handling large-scale TableQA tasks.

## 4.5 Retrieval Performance Analysis

To better understand the retrieval quality of various table prompting approaches, in Table 3, we assessed the recall, precision and f1 score for the prompts fed to LMs for reasoning. The ground truths are extracted from the program annotations in the ArcadeQA and BirdQA datasets. In column retrieval, while all methods achieved high recall due to the small number of columns, TableRAG demonstrated superior precision across both datasets, indicating its effectiveness in identifying the most relevant columns concisely. In contrast, ReadSchema and RowColRetrieval showed lower precision, suggesting that they retrieved more irrelevant columns. For cell retrieval, TableRAG consistently outperformed other methods on all metrics. TableRAG's high recall in cell retrieval marks a significant improvement over other table prompting methods, indicating it can retrieve most necessary cells for the subsequent reasoning. In summary, this analysis underscores TableRAG's efficacy in retrieving essential information in both the column and cell aspects.

---

[3]https://mistral.ai/news/mistral-nemo/
[4]https://openai.com/index/new-embedding-models-and-api-updates/

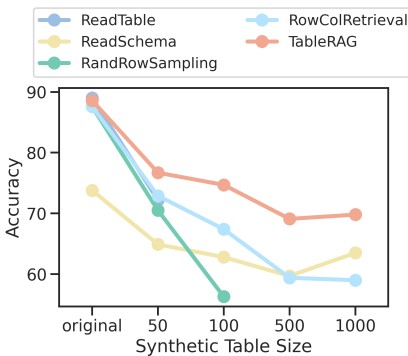

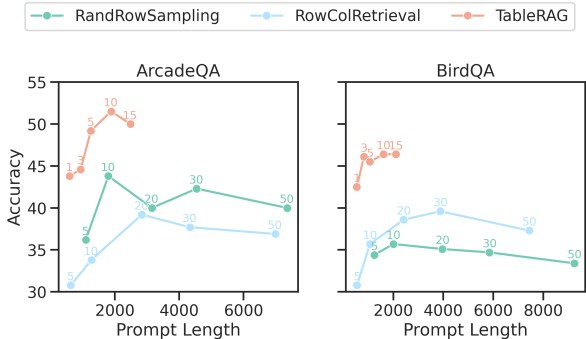

Figure 4: Performance evaluation of Synthetic Tabfact in varying scales. TableRAG shows consistently superior results, and it decreases gracefully compared to competitive methods.

Figure 5: Impact of varying top retrieval results ($K$). Different $K$ values influence both prompt length and accuracy. Each point is labeled with its corresponding $K$ value. TableRAG retrieves the top $K$ schema and cell values, RandRowSampling selects $K$ random rows, and RowColRetrieval retrieves $K$ rows and $K$ columns.

Table 4: Comparison of TableRAG with state-of-the-art methods on WikiTableQA.

| Method | Accuracy |
|---|---|
| TaBERT [25] | 52.30 |
| Text-to-SQL [15] | 52.90 |
| Binder [3] | 56.74 |
| Dater [24] | 52.81 |
| **TableRAG (Ours)** | **57.03** |

Table 5: Performance comparison of different retrieval approaches in TableRAG on ArcadeQA and BirdQA.

| Method | ArcadeQA | BirdQA |
|---|---|---|
| TableRAG (BM25) | 37.7 | 35.7 |
| TableRAG (Hybrid) | 46.2 | 44.5 |
| **TableRAG (Embed)** | **49.2** | **45.5** |

### 4.6 Scalability Test on TabFact

To understand the performance across varying table sizes under similar conditions, we create a set of synthetic data from TabFact with table sizes ranging from $50 \times 50$, $100 \times 100$, $500 \times 500$, and $1000 \times 1000$. The synthetic data allow us to analysis how table prompting methods perform in different scales with the same questions and key table contents. The results are shown in Fig. 4. ReadTable exhibited strong initial accuracy with the original data but failed dramatically as table sizes increased, infeasible for sizes 100 and beyond due to the context length limitations. Conversely, TableRAG demonstrated the most consistent and scalable performance, decreasing moderately from 83.1 to 68.4 as table size increased to 1000 rows and columns, showcasing its effectiveness in understanding larger tables. Both ReadSchema and RowColRetrieval showed declines in performance with increasing table size, yet maintained moderate accuracy, highlighting their relative scalability compared to ReadTable but less effectiveness than TableRAG in handling large tables.

### 4.7 Comparison with State-of-the-Art on WikiTableQA

To evaluate performance on small-scale TableQA datasets, we compared TableRAG with state-of-the-art approaches that rely on reading entire tables, using the commonly used WikiTableQA [12] benchmark. As shown in Table 4, TableRAG surpasses all existing methods, including TaBERT [25], Text-to-SQL [15], Binder [3], and Dater [24]. This highlights TableRAG's effectiveness, even in small-scale settings. These results confirm that, while TableRAG is designed for large-scale TableQA, its approach is versatile and maintains state-of-the-art performance across different table sizes and complexities.

### 4.8 Ablation Studies

**Impact of Retrieval Methods in TableRAG:** Table 5 compares different retrieval approaches within TableRAG. BM25 [16], a well-known statistical retrieval method, excels in efficiency and can process all cells but lacks semantic understanding. We also compare it with our embedding-based retrieval and

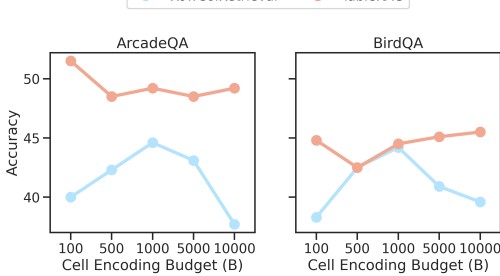

Figure 6: Impact of cell encoding budget $B$. TableRAG retrieves from the $B$ most frequent cells, maintaining robust performance even with a smaller budget. RowColSampling truncates more rows as $B$ decreases, showing greater sensitivity to budget changes and generally underperforming compared to TableRAG.

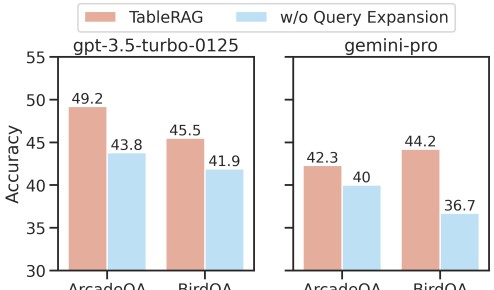

Figure 7: Effect of query expansion. Query expansion consistently improves TableRAG across all scenarios, indicating that it provides better coverage of user intents.

Table 6: Ablation study of schema retrieval (Rows 1 vs 3 and 2 vs 4) and cell retrieval (Rows 1 vs 2 and 3 vs 4). The first column indicates whether the LM processed all schemas or only retrieved schemas. The second column indicates whether the LM ignored cell values or processed retrieved column-cell pairs. Schema retrieval leads to an improvement in accuracy of up to 9.4%, while cell retrieval results in an increase of up to 11.5%.

| Schema | Cell | GPT 3.5 Turbo | | Gemini 1.0 Pro | |
|---|---|---|---|---|---|
| | | ArcadeQA | BirdQA | ArcadeQA | BirdQA |
| all | none | 43.1 | 40.3 | 30.8 | 31.8 |
| all | retrieval | 48.5 | 42.5 | 42.3 | 41.6 |
| retrieval | none | 46.9 | 44.8 | 36.9 | 42.2 |
| retrieval | retrieval | 49.2 | 45.5 | 42.3 | 44.2 |

a hybrid approach that combines scores from both methods. The results show that embedding-based retrieval achieves the best performance, outperforming both BM25 and the hybrid method, despite not processing the entire table due to encoding constraints. This underscores the importance of semantic understanding in retrieval, where embedding-based methods offer better comprehension of table data, significantly enhancing TableRAG's performance.

**Number of Top Retrieval Results $K$:** Fig 5 illustrates the impact of varying the number of top retrieval results ($K$) on the performance and the token cost for the subsequent LM reasoning. The results demonstrate that that increasing the number $K$, while increasing the prompt lengths, does not consistently improve performance. Though larger $K$ allows LMs access to more information, it also results in a longer context which can exacerbate the lost-in-the-middle phenomenon. In contrast, TableRAG excels by requiring fewer $K$ values, thus reducing the context tokens needed and lowering subsequent reasoning costs while still outperforming other methods.

**Encoding Budget $B$:** The results from Fig. 6 demonstrate how different token encoding budgets ($B$) affect the performance of TableRAG and RowColRetrieval. While a higher budget theoretically allows for more information to be retrieved, the results show that it does not always lead to better performance. Specifically, RowColRetrieval shows a decline in performance with increased budgets, potentially due to the retrieval of more rows that complicate selecting the correct ones and produce noisier embeddings from longer column sequences. In contrast, TableRAG maintains consistent performance across various budgets, indicating that its approach of building the corpus by cell frequency effectively captures essential information even with limited budgets.

**Query Expansion:** The effectiveness of our query expansion method is analyzed in Fig. 7. The results demonstrate that query expansion consistently enhances TableRAG's performance across various datasets and LMs.

**Schema Retrieval and Cell Retrieval:** We analyze the impact of schema retrieval and cell retrieval on performance in Table 6. The results demonstrate that schema retrieval consistently enhances reasoning performance across datasets and LMs, with a maximum improvement of $9.4\%$, regardless of whether cell values are considered. The results indicate that even for tables with small number of columns (average $20.5$ columns in ArcadeQA and $11.1$ columns in BirdQA), schema retrieval is still helpful to only provide relevant columns for the subsequent reasoning. Similarly, cell retrieval consistently improves accuracies across all datasets and LMs, with a maximum improvement of $11.5\%$, indicating cell retrieval can effectively extract key information from the table contents.

## 5 Conclusion

In this work, we have presented TableRAG, the first framework to demonstrate effective and efficient LM-based table understanding with millions of tokens. To fill in the evaluation gap, we have introduced three new million-token scale table understanding benchmarks. TableRAG's retrieval design, combining schema and cell retrieval with frequency-aware truncation, significantly reduces computational costs and token usage without sacrificing performance. Our extensive experiments on both real-world and synthetic datasets showcase TableRAG's superior performance across various table sizes. TableRAG paves the way for future research on even larger and more complex table understanding tasks.

## 6 Limitations

Theoretically, the worst-case complexity for cell retrieval could reach $O(NM)$ if all cell values are distinct, potentially leading to loss of crucial information even with frequency-based truncation. Also, our evaluation is limited to QA and verification tasks due to the scarcity of large-scale table benchmarks. Finally, we focus on benchmarking on comparing TableRAG with other representative table prompting methods using the same LM solver, rather than against the latest state-of-the-art techniques, to specifically assess its efficacy on large tables.

## Acknowledgement

S.-A. Chen and H.-T. Lin are partially supported by the National Taiwan University Center for Data Intelligence via NTU-113L900901 and the Ministry of Science and Technology in Taiwan via NSTC 113-2628-E-002-003.

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

# A  TableRAG Example

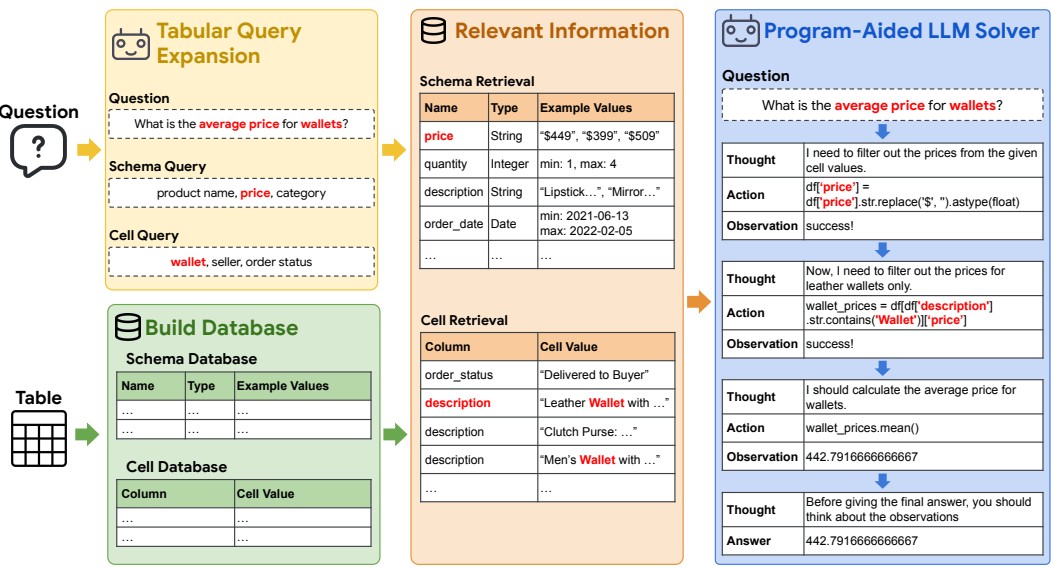

Figure 8: Example of TableRAG in ArcadeQA.

## B  TableRAG Pseudo Code

See Alg. 1.

---

**Algorithm 1:** TableRAG Algorithm

**Data:** $T$: table, $Q$: questions, $\mathcal{L}$: LM, $K$: number of retrieval results, $B$: token encoding budget, $f_{\text{enc}}$: encoder

**Result:** $\hat{A}$ is the predicted answer to the question

1 **Function** TableRAG($T, Q, \mathcal{L}, K, B, f_{enc}$)**:**
2     schema_db $\leftarrow$ BuildSchemaDB($T, f_{\text{enc}}$);
3     cell_db $\leftarrow$ BuildCellDB($T, B, f_{\text{enc}}$);
4     **foreach** $q \in Q$ **do**
5        $Q_{\text{schema}}, Q_{\text{cell}} \leftarrow$ TabularQueryExpansion($\mathcal{L}, q$);
6        $R_{\text{schema}} \leftarrow$ MultiQueryRetrieve($f_{\text{enc}}, Q_{\text{schema}}, K,$ schema_db);
7        $R_{\text{cell}} \leftarrow$ MultiQueryRetrieve($f_{\text{enc}}, Q_{\text{cell}}, K,$ cell_db);
8        $\hat{A} \leftarrow$ ProgramAidedSolver($\mathcal{L}, q, T, R_{\text{schema}}, R_{\text{cell}}$);
9     **end**
10 **return** $\hat{A}$
11 **Function** BuildSchemaDB($T, f_{enc}$)**:**
12     corpus $\leftarrow \phi$;
13     **foreach** $(C_j, D_j, E_j) \in T$ **do**
14        corpus $\leftarrow$ corpus $\cup (f_{Enc}(C_j), (C_j, D_j, E_j))$;
15     **end**
16 **return corpus**
17 **Function** BuildCellDB($T, B, f_{enc}$)**:**
18     corpus $\leftarrow \phi$;
19     $V \leftarrow$ getDistinctColumnCellPairbyFreq($T$);
20     $V \leftarrow V[:B]$;
21     **foreach** $(C_j, v_{ij}) \in V$ **do**
22        corpus $\leftarrow$ corpus $\cup (f_{Enc}((C_j, v_{ij})), (C_j, v_{ij}))$
23     **end**
24 **return corpus**
25 **Function** MultiQueryRetrieve($\mathcal{L}, Q, K,$ *corpus*)**:**
26     $res \leftarrow \phi$;
27     **foreach** $q \in Q$ **do**
28        $res \leftarrow res \cup$ Retrieve$_K(q,$ corpus$)$
29     **end**
30 **return** $res$

---

# C  Dataset

Here we describe the generation of each datasets. The statistics of each dataset are shown in Table 7.

**ArcadeQA**  The ARCADE dataset [26] consists of a collection of annotated data science notebooks with natural language to code annotations for every cell. The notebooks executes on large scale Kaggle ML datasets[5]. We extract the examples where there is an executed result, and that result is either a single primitive value or a list of values (single column table). In contrast, examples that focus on data wrangling without a returned value, or where the result is a table are skipped. In total we obtain 130 examples with questions such as *"How many countries are south of the equator?"* and *"What is the city that had cash on delivery as the most common payment method?"*.

**BirdQA**  We leverage a subset of BIRD dataset [7], a collection for large-scale database grounded text-to-SQL. It also contains annotations for columns descriptions, and external knowledge evidence in some examples. To convert the task to table-QA, we extract 308 examples where a single table was used. We use only the table and question without additional helper annotations such as descriptions and external knowledge evidence. The tables have 40 thousand rows in average and it comprises on single an multiple value answers.

**Synthetic TabFact**  To complement existing datasets and better evaluate LMs on larger table structures, we have synthesized an extension of the TabFact dataset to include tables of significantly greater sizes. This synthesis involves a series of steps designed to expand original small tables into larger ones without losing the integrity of the data or the logical consistency required for correct query answering. We utilize LM-generated column names, sampling functions, and solution programs to create tables and validate them through a rigorous process where solution programs verify the correctness of answers, ensuring the synthetic tables maintain the challenge and integrity of the original dataset.

Table 7: Dataset statistics. Values are presented in averages, except for the total counts of tables and questions.

| Data | # of tables | # of questions | # of rows $(N)$ | # of cols $(M)$ | # of cells $(NM)$ | # of distinct values $(D)$ | # of categorical columns |
|---|---|---|---|---|---|---|---|
| ArcadeQA | 48 | 130 | 79,376.2 | 20.5 | 1,247,946.6 | 50,609.4 | 9.4 |
| BirdQA | 53 | 308 | 62,813.1 | 11.1 | 721,077.6 | 39,649.5 | 4.9 |
| Synth-TabFact-original | 288 | 288 | 14.1 | 6.3 | 88.8 | 41.1 | 4.4 |
| Synth-TabFact-50 | 288 | 288 | 50.0 | 50.0 | 2,500.0 | 149.1 | 27.0 |
| Synth-TabFact-100 | 288 | 288 | 100.0 | 100.0 | 10,000.0 | 268.1 | 54.1 |
| Synth-TabFact-500 | 288 | 288 | 500.0 | 500.0 | 250,000.0 | 1,140.5 | 267.2 |
| Synth-TabFact-1K | 288 | 288 | 1,000.0 | 1,000.0 | 1,000,000.0 | 2,196.8 | 539.9 |

---

[5]https://www.kaggle.com/datasets

## D  Prompt of Query Expansion for Schema Retrieval

```
==================================== Prompt ====================================

Given a large table regarding "amazon seller order status prediction orders data", I want
to answer a question: "What is the average price for leather wallets?"
Since I cannot view the table directly, please suggest some column names that might contain
 the necessary data to answer this question.
Please answer with a list of column names in JSON format without any additional explanation
.
Example:
["column1", "column2", "column3"]

================================== Completion ==================================
["product_name", "category", "price"]
```

Figure 9: Prompt of Query Expansion for Schema Retrieval

## E  Prompt of Query Expansion for Cell Retrieval

```
==================================== Prompt ====================================

Given a large table regarding "amazon seller order status prediction orders data", I want
to answer a question: "What is the average price for leather wallets?"
Please extract some keywords which might appear in the table cells and help answer the
question.
The keywords should be categorical values rather than numerical values.
The keywords should be contained in the question.
Please answer with a list of keywords in JSON format without any additional explanation.
Example:
["keyword1", "keyword2", "keyword3"]

================================== Completion ==================================
["leather wallets", "average price", "amazon seller", "order status prediction", "orders
data"]
```

Figure 10: Prompt of Query Expansion for Cell Retrieval

# F Prompt of TableRAG Solver

```
===================================== Prompt =====================================

You are working with a pandas dataframe regarding "amazon seller order status prediction
orders data" in Python. The name of the dataframe is 'df'. Your task is to use '
python_repl_ast' to answer the question: "What is the average price for leather wallets?"

Tool description:
- 'python_repl_ast': A Python interactive shell. Use this to execute python commands. Input
  should be a valid single line python command.

Since you cannot view the table directly, here are some schemas and cell values retrieved
from the table.

Schema Retrieval Results:
{"column_name": "item_total", "dtype": "object", "cell_examples": ['$449.00', '$399.00', '
$549.00']]}
{"column_name": "quantity", "dtype": "int64", "min": 1, "max": 4}
{"column_name": "order_no", "dtype": "object", "cell_examples": ['402-4845680-8041921',
'405-9763961-5211537', '404-3964908-7850720']]}
...

Cell Retrieval Queries: leather wallets, average price, order status, prediction, amazon
seller
Cell Retrieval Results:
{"column_name": "order_status", "cell_value": "Delivered to buyer"}
{"column_name": "description", "cell_value": "Pure Leather Camel Color Gent's Wallet with
Coin Compartment and Card Holders | Men's Ultra Slim Money Organiser (1 pc)"}
...

Strictly follow the given format to respond:
Thought: you should always think about what to do
Action: the single line Python command to execute
Observation: the result of the action
... (this Thought/Action/Observation can repeat N times)
Thought: before giving the final answer, you should think about the observations
Final Answer: the final answer to the original input question (Answer1, Answer2, ...)

Notes:
- Do not use markdown or any other formatting in your responses.
- Ensure the last line is only "Final Answer: Answer1, Answer2, ..." form, no other form.
- Directly output the Final Answer rather than outputting by Python.
- Ensure to have a concluding thought that verifies the table, observations and the
question before giving the final answer.

Now, given a table regarding "amazon seller order status prediction orders data", please
use 'python_repl_ast' with the column names and cell values above to answer the question: "
What is the average price for leather wallets?"

Begin!

=================================== Completion ===================================
Thought: I need to remove the comma in the item_total values before converting them to
float.
Action: df['item_total'] = df['item_total'].str.replace('$','').str.replace(',','').astype(
float)
Observation: success!
Thought: Now, I can filter out the rows with leather wallets and calculate the average
price.
Action: df[df['description'].str.contains('wallet', case=False)]['item_total'].mean()
Observation: 442.7916666666667
Thought: before giving the final answer, I should consider the observations and ensure that
 the calculations are accurate.
Final Answer: The average price for leather wallets is $442.79.
```

Figure 11: Prompt of TableRAG solver

