# OpenReview forum: "TableRAG: Million-Token Table Understanding with Language Models"
_NeurIPS.cc/2024/Conference — NeurIPS 2024 poster_

### Official Review · Reviewer_aMG9 · 2024-07-05

**Soundness:** 3
**Presentation:** 3
**Contribution:** 2
**Rating:** 5
**Confidence:** 4

**Summary:**

This paper primarily concentrates on the scalability challenges associated with encoding entire tables as input for LLM reasoning. It introduces a retrieval-augmented generation (RAG) framework, named TableRAG, which utilizes query expansion along with schema and cell retrieval to identify essential information effectively. Furthermore, this study contributes to establishing new benchmarks for million-token-sized tables, significantly enhancing the understanding of LLM performance in large-scale table analysis.

**Strengths:**

- the idea of schema-cell retrieval seems reasonable and shows better performance compared to baselines on some types of questions.
- this paper seems contribute to new benchmarks concerning million-token sized tables and enhances the understanding of LLM performance in large-scale table comprehension.

**Weaknesses:**

- while the idea of schema-cell retrieval seems reasonable, the generalizability of the method is not very clear, especially for some questions that cannot be directly addressed by manipulating the tables (also noted in the limitation section)
- the experiments only consider GPT-3.5-turbo and Gemini-Pro as LLMs which are limited, further testing on more open-sourced and close-sourced models is necessary to prove the consistency of the proposed method.
- the performance in Table 4 appears effective solely for the ArcadeQA dataset, with the retrieval of schema and cells proving useful. However, for BirdQA, the enhancements seem limited. I'm little concerned about the method consistency when adding more datasets.

**Questions:**

- for figure 8, I am slightly confused by the results obtained from the solver. It appears the query intends to request the average price for all types of wallets. Yet, in the provided example, the third action only yields the average price of leather wallets (leather_wallet_prices.means()), rather than the average for all wallets (wallet_prices.means()). I'm uncertain if this discrepancy stems from a misrepresentation or if there's potential confusion in the example.
- for controlling the budget in cell retrieval, how does the paper manage a column like "description," which consists of long sentences that might exhaust the entire budget, preventing other cells from being processed?
- for schema/cell retrieval, if the query expansion driven by LLMs fails to generate keywords that align with the schema/cells in the table, what will the outcomes be, or are there any solutions proposed in the paper? Additionally, I am curious about how the paper attempts to align the schema mentioned in the table with the knowledge from LLMs themselves, or does it simply rely on the generalizability of LLMs?

**Limitations:**

- the proposed method seems to primarily support the questions that can be answered by manipulating the tables, such as "What is the average price of the wallet?" Here, queries can be transformed into specific table operations (e.g., df['item_total'] = df['item_total'].str.replace('$','').str.replace(',','').astype(float)). However, it may not perform as well with more complex reasoning tasks. For instance, in the Tabfact dataset, where one must determine from a Wikipedia table and its caption which statements are supported or contradicted, the method might not show promising results for such reasoning tasks.

---

> ### Author Rebuttal · Authors · 2024-08-07
>
> ## Generalizability
>
> We appreciate the reviewer's question regarding the generalizability of our schema-cell retrieval method. It's important to note that our work focuses on general TableQA, which often involves complex understanding and reasoning beyond simple table manipulation, as shown in the TabFact example in the general response. This distinguishes our approach from Text2SQL methods, which have a narrower scope.
>
> The generalizability of TableRAG is inherently tied to the capabilities of the underlying solver agent. TableRAG can be easily integrated with any downstream LLM-based solver agents. In our implementation, we utilize ReAct, a powerful LLM-based solver with demonstrated effectiveness and generalizability in TableQA tasks [1, 2]. TableRAG's primary role is to efficiently and accurately provide the necessary information to the solver, enabling it to perform the required reasoning and extract relevant details.
>
> 1. [Rethinking Tabular Data Understanding with Large Language Models](https://aclanthology.org/2024.naacl-long.26) (Liu et al., NAACL 2024)
> 2. [ReAcTable: Enhancing ReAct for Table Question Answering](https://dl.acm.org/doi/10.14778/3659437.3659452)  (Zhang et al., VLDB 2024)
>
> ## Evaluation on more models
>
> We follow reviewer’s suggestion and include results obtained using Mistral Nemo, as shown in the general response.
> The results demonstrate that TableRAG remains the most effective table prompting technique when working with Mistral Nemo, consistent with our previous findings using GPT 3.5 and Gemini 1.0 Pro.
>
> ## Ablation study
>
> Thanks for the insightful feedback. The accuracy improvement on BirdQA ranges from 0.7% to 2%, which is in line with the improvements observed in prior work such as Binder[3] and Dater[4].
>
> We observe an even more substantial accuracy boost on ArcadeQA, ranging from 2.3% to 6.6%. This is likely attributable to the larger scale of the input in this dataset. As detailed in Table 5, ArcadeQA has approximately twice the average number of columns and cells compared to BirdQA, suggesting that cell retrieval is particularly beneficial when handling larger tables. Nonetheless, both schema and cell retrieval components enhance performance across all tested datasets, and they contribute orthogonal advantages to the overall improvement.
>
> 3. [Binding Language Models in Symbolic Languages](https://openreview.net/forum?id=lH1PV42cbF) (Cheng et al., ICLR 2023)
> 4. [Large Language Models are Versatile Decomposers: Decomposing Evidence and Questions for Table-based Reasoning](https://dl.acm.org/doi/10.1145/3539618.3591708) (Ye et al., SIGIR 2024)
>
> ## Figure 8
>
> Thanks for pointing out the typo. The variable `leather_wallet_prices` should indeed be `wallet_prices` as obtained in the previous step. We will correct this in the final version.
>
> ## Description column
>
> Thank you for raising this important point about handling long text columns within our encoding budget. We acknowledge this limitation and designed TableRAG to mitigate its impact in the following ways:
> 1. **Prioritization of Categorical Columns**: Our frequency-based encoding prioritizes columns with more distinct categorical values, ensuring that critical information for indexing is processed first. This helps reduce the likelihood of long text columns exhausting the entire budget.
> 2. **Solver Agent's String Matching Capability**: Even if the retrieval doesn't cover all cells in long text columns like "description" or "address," our solver agent (ReAct in our case) can still perform basic string matching within those columns to identify relevant information. As demonstrated in Figure 8, the solver agent can detect the existence of the “description” column through retrieval and then use basic string matching techniques (e.g., `df['description'] .str.contains('Wallet')`) to pinpoint the precise cell within those columns.
> 3. **Empirical Evidence from the Study of Encoding Budget**: The ablation study in Figure 6 shows that TableRAG is robust to changes in the encoding budget. We varied the encoding budget from 100 to 10,000 and found that TableRAG's performance remained near optimal on both ArcadeQA and BirdQA. In contrast, Row-Column Retrieval is sensitive to the encoding budget, with performance significantly dropping when the encoding budget is either increased or decreased. This study validates the earlier statement that even if not all relevant cells are retrieved, the solver agent can still obtain the necessary information through programs using the retrieved information in most cases.
>
> While we recognize that there might be more optimized ways to handle long text columns, our current work prioritizes the implementation and evaluation of query expansion and schema-cell retrieval, which we believe are foundational for scaling LM-based TableQA. Addressing the specific challenge of long text columns will be a focus of our future research.
>
> ## False alignment
> Thank you for raising these important questions about the robustness of our retrieval and the alignment between schema and LLM knowledge.
>
> **Handling Query Expansion Mismatches**: In TableRAG, we employ embedding-based retrieval, which inherently captures semantic similarities. Even if the LLM-generated keywords don't perfectly match the schema or cell values, the retrieval process will still return the most semantically relevant ones. This provides a degree of robustness against potential mismatches in query expansion.
>
> **Schema Alignment and LLM Generalizability**: TableRAG, having access to both the query and the retrieved context, is expected to leverage its understanding of language and world knowledge to determine which information is pertinent and how to combine it to solve the problem at hand. While we don't explicitly enforce schema alignment within the LLM, we rely on its inherent ability to reason and connect information, even when the schema terms might not have been directly encountered during pre-training.

---

> > ### Comment · Reviewer_aMG9 · 2024-08-08
> >
> > Thank you for addressing my concerns. I'm now willing to increase my rating from 4 to 5. I think the contributions are clear, and reasonable, including (1) schema-cell retrieval and (2) new benchmark considering million-token size tables. Cheers!

---

> > > ### Author Response · Authors · 2024-08-08
> > >
> > > Thank you for recognizing the contribution of this work and increasing the review rating! We appreciate your valuable feedback and will incorporate it into the final version.

---

### Official Review · Reviewer_6sEk · 2024-07-06

**Soundness:** 3
**Presentation:** 3
**Contribution:** 2
**Rating:** 5
**Confidence:** 4

**Summary:**

The paper introduces TableRAG, a framework that enhances LM-based table understanding by incorporating query expansion and retrieval mechanisms. TableRAG aims to improve the performance of large-scale table understanding tasks by efficiently encoding data and utilizing precise retrieval techniques. The framework achieves state-of-the-art results on various benchmarks and datasets.

**Strengths:**

1. The overall method is relatively simple and intuitive.
2. Experimental results on three datasets show the effectiveness of the proposed method.

**Weaknesses:**

1. The contribution of the paper is relatively small. The author mainly builds upon the original Row Column Retrieval and further reduces the retrieval cost by Scheme Cell Retrieval. The original work has included column selection, but this paper only further reduces the returned row to only return specific cells.
2. The baseline of the experimental comparison is relatively weak, and the advantage of finer grained retrieval is obvious.

**Questions:**

N/A

---

> ### Author Rebuttal · Authors · 2024-08-07
>
> ## Contribution
> > The contribution of the paper is relatively small. The author mainly builds upon the original Row Column Retrieval and further reduces the retrieval cost by Scheme Cell Retrieval. The original work has included column selection, but this paper only further reduces the returned row to only return specific cells.
>
> We thank the reviewer for the constructive feedback. We would like to emphasize that the existing Row-Column Retrieval requires encoding all rows and columns, which is infeasible for large-scale tables without significant truncation. Our work directly addresses this critical limitation and is significantly differentiated in 3 aspects:
> 1. **New Benchmarks**: We introduce two real-world large-scale TableQA benchmarks, ArcadeQA and BirdQA, filling a critical gap in the field. These benchmarks enable rigorous evaluation of TableQA systems on tables with millions of tokens, pushing the boundaries of research in this domain.
> 2. **Scalable Framework**: We propose TableRAG, a comprehensive framework that combines novel techniques such as query expansion, frequency-based filtering, and schema-cell retrieval. This framework enables LM-based TableQA to scale to tables with millions of tokens, a significant advancement over existing methods.
> 3. **Rigorous Evaluation**: We provide thorough complexity analysis and extensive empirical studies, demonstrating TableRAG's effectiveness and efficiency. Our work paves the way for broader applications of TableQA in real-world scenarios involving large tables.
>
> We will make these clearer in the final version.
>
>
> ## Baselines
> > The baseline of the experimental comparison is relatively weak, and the advantage of finer grained retrieval is obvious.
>
> To the best of our knowledge, TableRAG represents the first dedicated effort to address LM-based large-scale table understanding. We are eager to include any specific baselines the reviewer may recommend that we might have overlooked. While we acknowledge the effectiveness of existing methods on smaller tables, our analysis and studies clearly demonstrate their limitations in scaling due to context length constraints. Furthermore, we have shown that previous attempts at scalable retrieval methods for tabular data encountered inherent limitations. Our carefully designed TableRAG framework overcomes these challenges, achieving superior performance and scalability, as validated through comprehensive complexity analysis and extensive ablation studies.
>
> To better understand TableRAG’s performance compared against state-of-the-art baselines, perform experiments on WikiTableQA with GPT 3.5 as shown below:
>
> | Method                              |  Accuracy |
> |-------------------------------------|:---------:|
> | TaBERT (Yin et al., 2020)           |    52.3   |
> | Text-to-SQL (Rajkumar et al., 2022) |    52.9   |
> | Binder (Cheng et al., 2022)         |   56.74   |
> | Dater (Ye et al., 2023)             |   52.81   |
> | **TableRAG (Ours)**                 | **57.03** |
>
> In this benchmark, the entire table content often fits within the context window of the language models, which is not representative of real-world large-scale industrial scenarios.
>
> Nevertheless, the results clearly indicate that TableRAG maintains its superior performance even on smaller tables. It's important to highlight that the baseline methods require that LLMs process the entire table, thus limiting their scalability for large tables.

---

> > ### Author Response · Authors · 2024-08-12
> >
> > We've submitted our rebuttal, and Reviewer aMG9 has already found it helpful enough to increase their rating.  We're hoping you might also get a chance to check it out before the deadline in two days – we'd love to hear your thoughts!

---

> > > ### Author Response · Authors · 2024-08-13
> > >
> > > Dear reviewer, we're hoping you might get a chance to check out the rebuttal before the deadline today. We'd love to get your feedback!

---

> > > > ### Comment · Reviewer_6sEk · 2024-08-14
> > > >
> > > > Thank you for the response! For the concern about the novelty of the paper, I keep my rating.

---

### Official Review · Reviewer_yX7T · 2024-07-07

**Soundness:** 3
**Presentation:** 3
**Contribution:** 2
**Rating:** 4
**Confidence:** 4

**Summary:**

This paper introduces a novel approach to retrieving cell values and database schema within the table question answering domain. The method operates as follows:
1) It expands the initial question by identifying potential queries for column retrieval or cell value retrieval.
2) For each query generated in the first step, the method identifies the top-K columns using their embeddings.
3) Within a given encoding budget for cell retrieval, it proposes embedding the most frequent B distinct values for each column and retrieves them based on their similarity to each query.
Additionally, the authors have developed two new datasets for large-scale table question answering. Their approach demonstrates improved cost efficiency and effectiveness on these large-scale datasets compared to previous methods.

**Strengths:**

1) The proposed approach significantly outperforms the baseline models, particularly in retrieving cell values, showing a substantial improvement in recall metric.

2) Unlike previous works in the table question answering domain, which focused on small-scale tables that do not accurately represent real-world databases, this approach tackles a more challenging setting. It addresses scenarios where tables can contain millions of rows and numerous columns. The authors also developed two new datasets specifically designed for large-scale table question answering.

3) The paper includes well-conducted ablation studies and experiments that compare their method against their baseline methods. These studies demonstrate the effectiveness of each component of their approach.

**Weaknesses:**

1) My primary concern pertains to the comprehensiveness of the baselines used. While the paper correctly mentions that the table schema is predominantly considered in the text-to-SQL domain, cell values also play a crucial role in SQL generation. Notably, papers like CodeS [1] have proposed using BM25 retrieval to find cell values, which would serve as an excellent baseline for comparison. Additionally, schema linking—the process of identifying the correct rows and columns—is well-explored. Approaches proposed in studies like TaBERT [2] could also be utilized as baselines.

2) Given an encoding budget, the strategy of retaining only the most frequent distinct cell values seems problematic, especially for queries that require data from columns containing names, addresses, etc. Heuristic-based methods, such as those using syntactic similarity with edit distance, can effectively filter most of the cell values and might be more appropriate.

[1]: CodeS: Towards Building Open-source Language Models for Text-to-SQL
[2]: TaBERT: Pretraining for Joint Understanding of Textual and Tabular Data

**Questions:**

Not applicable

---

> ### Author Rebuttal · Authors · 2024-08-07
>
> ## Baselines
> > Papers like CodeS [1] have proposed using BM25 retrieval to find cell values, which would serve as an excellent baseline for comparison. Additionally, schema linking—the process of identifying the correct rows and columns—is well-explored. Approaches proposed in studies like TaBERT [2] could also be utilized as baselines.
>
> We thank the reviewer for the valuable references. It's important to note that our work focuses on general TableQA, which often involves complex understanding and reasoning beyond simple table manipulation. This distinguishes our approach from Text2SQL methods, which have a narrower scope. While CodeS primarily focuses on Text2SQL, its BM25 retrieval approach offers a potential enhancement to our TableRAG framework. Thus, we implement BM25 and [hybrid retrieval](https://python.langchain.com/v0.1/docs/modules/data_connection/retrievers/ensemble/) as a replacement for embedding-based retrieval to better understand the impact of different retrieval methods.
>
> | Method               |          | ArcadeQA |          |          |  BirdQA  |          |
> |----------------------|:--------:|:--------:|:--------:|:--------:|:--------:|:--------:|
> |                      |   small  |   large  |   full   |   small  |   large  |   full   |
> | ReadTable            |   18.2   |    0.0   |    4.6   |   36.4   |    0.0   |    9.1   |
> | ReadSchema           |   48.5   |   41.2   |   43.1   |   49.4   |   37.2   |   40.3   |
> | RandRowSampling      |   42.4   |   40.2   |   42.3   |   49.4   |   29.9   |   34.7   |
> | RowColRetrieval      |   39.4   |   37.1   |   37.7   |   49.4   |   36.4   |   39.6   |
> | TableRAG (BM25)      |   45.5   |   35.1   |   37.7   |   46.8   |   32.0   |   35.7   |
> | TableRAG (Hybrid)    |   51.5   |   46.4   |   46.2   |   54.5   |   41.1   |   44.5   |
> | **TableRAG (Embed)** | **54.5** | **47.4** | **49.2** | **55.8** | **42.0** | **45.5** |
>
> The results demonstrate that TableRAG with embedding-based retrieval still outperforms BM25 and hybrid retrieval consistently.
>
> Regarding TaBERT and other LM-based methods, their effectiveness is indeed constrained by the context window of language models, making them less suitable for large-scale tables. Nevertheless, we recognize the benefit of comparing TableRAG against representative baselines, and we evaluate TableRAG on additional WikiTableQA benchmark  with GPT 3.5 as shown below:
>
> | Method                              |  Accuracy |
> |-------------------------------------|:---------:|
> | TaBERT (Yin et al., 2020)           |    52.3   |
> | Text-to-SQL (Rajkumar et al., 2022) |    52.9   |
> | Binder (Cheng et al., 2022)         |   56.74   |
> | Dater (Ye et al., 2023)             |   52.81   |
> | **TableRAG (Ours)**                 | **57.03** |
>
> In this benchmark, the entire table content often fits within the context window of the language models, which is not representative of real-world large-scale industrial scenarios.
>
> Nevertheless, the results clearly indicate that TableRAG maintains its superior performance even on smaller tables. It's important to highlight that the baseline methods require that LLMs process the entire table, thus limiting their scalability for large tables.
>
>
> ## Limitation of retaining most frequent distinct cell values
> > Given an encoding budget, the strategy of retaining only the most frequent distinct cell values seems problematic, especially for queries that require data from columns containing names, addresses, etc. Heuristic-based methods, such as those using syntactic similarity with edit distance, can effectively filter most of the cell values and might be more appropriate.
>
> We appreciate the reviewer highlighting this potential limitation of the cell value filtering strategy. We were indeed aware of this trade-off during the design process and opted for LM-based retrieval with an encoding budget for the following reasons:
> 1. **Semantic Understanding vs. Speed**: While heuristic-based methods offer computational efficiency, they often fail to capture subtle semantic relationships that are crucial for accurate retrieval in complex TableQA tasks. We have presented results from replacing embedding-based retrieval with BM25 and a hybrid approach above. While BM25 and the hybrid approach excel at retrieving all cells from the table, their inferior semantic understanding capabilities result in poorer overall performance.
> 2. **Retrieval as Guidance, not the Final Answer**: Even with a limited encoding budget, our retrieval mechanism effectively guides the solver agent (PyReAct in our case) to the relevant columns. As demonstrated in Figure 8, the solver agent can detect the existence of the “description” column through retrieval and then use basic string matching techniques (e.g., `df['description'] .str.contains('Wallet')`) to pinpoint the precise cell within those columns. Thus, while retrieval may not always return all relevant cells, it remains crucial for directing the solver's attention to the right areas of the table.
> 3. **Empirical Evidence from the Study of Encoding Budget**: The ablation study in Figure 6 shows that TableRAG is robust to changes in the encoding budget. We varied the encoding budget from 100 to 10,000 and found that TableRAG's performance remained near optimal on both ArcadeQA and BirdQA. In contrast, Row-Column Retrieval is sensitive to the encoding budget, with performance significantly dropping when the encoding budget is either increased or decreased. This study validates the earlier statement that even if not all relevant cells are retrieved, the solver agent can still obtain the necessary information through programs using the retrieved information in most cases.
>
> We will make these clearer in the final version.

---

> > ### Author Response · Authors · 2024-08-12
> >
> > We've submitted our rebuttal, and Reviewer aMG9 has already found it helpful enough to increase their rating.  We're hoping you might also get a chance to check it out before the deadline in two days – we'd love to hear your thoughts!

---

> > > ### Author Response · Authors · 2024-08-13
> > >
> > > Dear reviewer, we're hoping you might get a chance to check out the rebuttal before the deadline today. We'd love to get your feedback!

---

### Official Review · Reviewer_YK6m · 2024-07-11

**Soundness:** 2
**Presentation:** 3
**Contribution:** 3
**Rating:** 4
**Confidence:** 4

**Summary:**

The paper introduces TableRAG, a novel framework that improves LM-based table understanding by incorporating advanced query expansion and retrieval mechanisms. TableRAG addresses the critical scalability challenges associated with large-scale tables by efficiently encoding data and employing precise retrieval techniques. Through a comprehensive evaluation on self-constructed benchmarks sourced from real-world datasets and synthetic data from TabFact, TableRAG demonstrates superior performance and reduced token consumption across various table sizes.

**Strengths:**

1. **Scalability**: The framework is scalable for larger tables, as demonstrated by new benchmarks sourced from real-world datasets and synthetic data from TabFact.
2. **State-of-the-Art Performance**: TableRAG achieves the highest retrieval quality and new state-of-the-art performance on large-scale table understanding tasks.

**Weaknesses:**

1. **Lack of Novelty** Many papers about RAG reveals that the accuracy of the evidence is crucially important for the performance, and the idea of this paper is to is another practice in the TableQA domain.
2. **Lack of evaluation robustness** The paper only evaluates the model on closed models gpt-3.5 and gemini, but fails to evaluate on open models like Mistral, Llama etc.

**Questions:**

1. Why not evaluate on more models?
2. Why not evaluation on commonly used TableQA benchmarks like WikiTableQA?

**Limitations:**

Yes, the paper discusses the limitations.

---

> ### Author Rebuttal · Authors · 2024-08-07
>
> ## Novelty
> >Many papers about RAG reveals that the accuracy of the evidence is crucially important for the performance, and the idea of this paper is to is another practice in the TableQA domain.
>
> Thank you for highlighting the importance of evidence accuracy in RAG systems. While this principle is well-established for unstructured data, its application to structured data like tables remains an open question. Prior retrieval-based methods for table understanding haven't consistently outperformed non-retrieval methods, primarily due to the challenges of efficient retrieval from tabular data. Our work addresses this gap by introducing a novel cell-schema retrieval method. Additionally, since existing benchmarks only focus on small tables, we contribute two large-scale TableQA datasets to facilitate further research in this area.
>
>
>
> ## Evaluation on open models
> >The paper only evaluates the model on closed models gpt-3.5 and gemini, but fails to evaluate on open models like Mistral, Llama etc.
> >Why not evaluate on more models?
> >Why not evaluation on commonly used TableQA benchmarks like WikiTableQA?
>
> We follow reviewer’s suggestion and include results obtained using Mistral Nemo, the latest model with a 128K context length from Mistral, as shown below.
>
> | Method          |          | ArcadeQA |          |          |  BirdQA  |          |
> |-----------------|:--------:|:--------:|:--------:|:--------:|:--------:|:--------:|
> |                 |   small  |   large  |   full   |   small  |   large  |   full   |
> | ReadTable       |   21.2   |    0.0   |    5.4   |   33.8   |    0.0   |    8.4   |
> | ReadSchema      |   36.4   |   30.9   |   32.3   |   41.6   |   33.8   |   35.7   |
> | RandRowSampling |   45.5   |   22.7   |   28.5   |   49.4   |   28.1   |   33.4   |
> | RowColRetrieval |   39.4   |   26.8   |   30.0   |   48.1   |   32.5   |   36.4   |
> | **TableRAG**    | **51.5** | **44.3** | **46.2** | **53.2** | **42.4** | **45.1** |
>
> The results demonstrate that TableRAG remains the most effective table prompting technique when working with Mistral Nemo, consistent with our previous findings using GPT 3.5 and Gemini 1.0 Pro. Furthermore, even with its 128K context length, Mistral Nemo struggles to handle large tables in ArcadeQA and BirdQA.
>
>
>
> ## Evaluation on WikiTableQA
> > Why not evaluation on commonly used TableQA benchmarks like WikiTableQA?
>
> In our paper, we have evaluated TabFact, a commonly used TableQA benchmark, and extended it to studying the behavior of TableRAG at different scales. In addition, we follow the reviewer's suggestion and perform experiments on WikiTableQA with GPT 3.5 as shown below:
>
> | Method                              |  Accuracy |
> |-------------------------------------|:---------:|
> | TaBERT (Yin et al., 2020)           |    52.3   |
> | Text-to-SQL (Rajkumar et al., 2022) |    52.9   |
> | Binder (Cheng et al., 2022)         |   56.74   |
> | Dater (Ye et al., 2023)             |   52.81   |
> | **TableRAG (Ours)**                 | **57.03** |
>
> In this benchmark, the entire table content often fits within the context window of the language models, which is not representative of real-world large-scale industrial scenarios.
>
> Nevertheless, the results clearly indicate that TableRAG maintains its superior performance even on smaller tables. It's important to highlight that the baseline methods require that LLMs process the entire table, thus limiting their scalability for large tables.

---

> > ### Author Response · Authors · 2024-08-12
> >
> > We've submitted our rebuttal, and Reviewer aMG9 has already found it helpful enough to increase their rating.  We're hoping you might also get a chance to check it out before the deadline in two days – we'd love to hear your thoughts!

---

> > > ### Comment · Reviewer_YK6m · 2024-08-12
> > >
> > > Thank you for the response! I still have concern about the novelty of the paper and it has not been addressed. I will keep my rating.

---

> > > > ### Author Response · Authors · 2024-08-13
> > > >
> > > > Thank you for reading our rebuttal and sharing your valuable feedback. We appreciate the opportunity to address the concerns regarding novelty. We believe the following points highlight the unique contributions and innovative aspects of our work:
> > > >
> > > > - While existing TableQA methods often focus on small tables, real-world applications frequently involve large and complex tables. **Even those using RAG techniques struggle to effectively encode entire large tables or resort to random truncation**, compromising accuracy. Our specialized RAG framework for table understanding directly tackles this challenge, offering a more **scalable** and accurate solution.
> > > > - Applying cell retrieval to TableQA is non-trivial. We address the difficulty of identifying relevant information without viewing the entire table through LM-based **query expansion** and **embedding-based retrieval**. This approach is supported by our ablation study (Figure 7) and further validated by additional experiments presented in our rebuttal.
> > > > - We also tackle the computational expense of encoding large tables by strategically encoding only distinct cells in categorical columns and representative cells in numerical columns, along with frequency-based filtering. Our empirical study (Figure 6) demonstrates that TableRAG maintains robust performance despite these optimizations, unlike other RAG methods that suffer from performance degradation due to random truncation.
> > > > - The benefits of cell retrieval extend beyond direct use. As shown in our general response, retrieved cells significantly aid the LLM in recognizing cell formats and identifying relevant columns that standard row/column retrieval methods might miss. This is especially valuable in complex tables, making our approach a substantial advancement for TableQA tasks.
> > > > - Current TableQA **benchmarks** primarily focus on **small tables**, limiting insights into real-world industrial scenarios. To address this, we introduce two new **large-scale** TableQA benchmarks, pushing the boundaries of the field and providing a deeper understanding of LLM capabilities on large tables.
> > > >
> > > > We firmly believe these points clearly demonstrate the novelty and significant impact of our work. Please let us know if we can provide further details for any of the specific points. Thanks!

---

### Author Rebuttal · Authors · 2024-08-07

## General Response

We thank the reviewers for their constructive feedback. They found our solution addresses the critical scalability challenge with superior performance in comprehensive evaluations. The work contributes to a better understanding of LLM capabilities in large-scale table analysis.

Our primary focus is to improve the scalability of LM-based table understanding. We achieve this through TableRAG, a novel table-specific retrieval technique incorporating query expansion, schema-cell retrieval, and frequency-based filtering. Additionally, we introduce ArcadeQA and BirdQA, two large-scale TableQA benchmarks specifically designed to evaluate and advance our understanding of LLM performance on large-scale tables.

We carefully address individual questions below and welcome further discussions of this work.

## Additional LLM evaluation

In addition to GPT 3.5 and Gemini 1.0 Pro, we have extended our evaluations to include Mistral Nemo, the latest model from Mistral with a 128K context window.

| Method          |          | ArcadeQA |          |          |  BirdQA  |          |
|-----------------|:--------:|:--------:|:--------:|:--------:|:--------:|:--------:|
|                 |   small  |   large  |   full   |   small  |   large  |   full   |
| ReadTable       |   21.2   |    0.0   |    5.4   |   33.8   |    0.0   |    8.4   |
| ReadSchema      |   36.4   |   30.9   |   32.3   |   41.6   |   33.8   |   35.7   |
| RandRowSampling |   45.5   |   22.7   |   28.5   |   49.4   |   28.1   |   33.4   |
| RowColRetrieval |   39.4   |   26.8   |   30.0   |   48.1   |   32.5   |   36.4   |
| **TableRAG**    | **51.5** | **44.3** | **46.2** | **53.2** | **42.4** | **45.1** |

- TableRAG outperforms all baselines, consistent with previous results observed in GPT 3.5 and Gemini 1.0 Pro.
- Despite their large 128K context window, LLMs continue to encounter challenges when dealing with large-scale tables. These challenges are largely mitigated by the proposed TableRAG


## Evaluation on common TableQA benchmark

We follow reviewer’s suggestion and evaluate TableRAG with GPT 3.5 on the additional WikiTableQA benchmark, comparing its performance against representative baselines:

| Method                              |  Accuracy |
|-------------------------------------|:---------:|
| TaBERT (Yin et al., 2020)           |    52.3   |
| Text-to-SQL (Rajkumar et al., 2022) |    52.9   |
| Binder (Cheng et al., 2022)         |   56.74   |
| Dater (Ye et al., 2023)             |   52.81   |
| **TableRAG (Ours)**                 | **57.03** |

- While TableRAG is primarily designed to address the challenges of large-scale tables, it also exhibits superior performance on smaller tables.


## Evaluation of Different Retrieval Methods

In response to the questions about the limited encoding budget of embedding-based retrieval methods, we follow the reviewer’s suggestion and implement BM25 and hybrid retrieval.
| Method               |          | ArcadeQA |          |          |  BirdQA  |          |
|----------------------|:--------:|:--------:|:--------:|:--------:|:--------:|:--------:|
|                      |   small  |   large  |   full   |   small  |   large  |   full   |
| ReadTable            |   18.2   |    0.0   |    4.6   |   36.4   |    0.0   |    9.1   |
| ReadSchema           |   48.5   |   41.2   |   43.1   |   49.4   |   37.2   |   40.3   |
| RandRowSampling      |   42.4   |   40.2   |   42.3   |   49.4   |   29.9   |   34.7   |
| RowColRetrieval      |   39.4   |   37.1   |   37.7   |   49.4   |   36.4   |   39.6   |
| TableRAG (BM25)      |   45.5   |   35.1   |   37.7   |   46.8   |   32.0   |   35.7   |
| TableRAG (Hybrid)    |   51.5   |   46.4   |   46.2   |   54.5   |   41.1   |   44.5   |
| **TableRAG (Embed)** | **54.5** | **47.4** | **49.2** | **55.8** | **42.0** | **45.5** |

- Although BM25 is faster and capable of retrieving all cells, its lack of semantic understanding makes embedding-based retrieval a superior choice for TableRAG. For a more detailed discussion, please refer to our response to Reviewer yX7T.

---

### Author Response · Authors · 2024-08-07
**TabFact Example**

To help reviewers better understand how TableRAG works and its capabilities, we provide a real example from TabFact for reference. When asked to verify, **“the game against New Orleans didn't have the highest attendance,”** TableRAG successfully answered the question with the following procedure:

> Schema Retrieval Queries: **opponent, attendance**
> Schema Retrieval Results:
> *{"column_name": "location attendance", "dtype": "object", "cell_examples": ['us airways center 18422', 'american airlines center 20301', 'new orleans arena 17781']}*
> … (omit other retrieval results)
>
> Cell Retrieval Queries: **2008-09 phoenix suns season, game against new orleans, highest attendance**
> Cell Retrieval Results:
> *{"column_name": "team", "cell_value": "new orleans"}*
> … (omit other retrieval results)
>
> Thought: **I need to find the attendance for the game against New Orleans and compare it with the attendance for other games.**
> Action: `df[df['team']=='new orleans']['location attendance']`
> *Observation:*
> *3    new orleans arena 17781*
> *Name: location attendance, dtype: object*
> Thought: **The attendance for the game against New Orleans was 17,781.**
> Action: `df['location attendance'].str.split(' ').str[-1].astype(int).max()`
> *Observation: ValueError: invalid literal for int() with base 10: 'center'*
> Thought: **I need to adjust the approach to extract and compare attendance values correctly.**
> Action: `df['location attendance'].str.split(' ').str[-1].str.replace('center', '').astype(int).max()`
> *Observation: ValueError: invalid literal for int() with base 10: ''*
> Thought: **I need to handle the case where the attendance value is not a valid integer.**
> Action: `df['location attendance'].str.extract('(\d+)').astype(float).max()`
> *Observation:*
> *0    20301.0*
> *dtype: float64*
> Thought: **The highest attendance in the table is 20,301, which is not for the game against New Orleans with an attendance of 17,781.
> Final Answer: Yes**

In this example, TableRAG performs the following steps:
1. Successfully retrieve relevant schema and cells.
2. Write the correct program to obtain the score against Team New Orleans.
3. Recognizes that the cell values are compound strings rather than numerical values.
4. Writes and corrects the program to parse the numerical values from the column and find the maximum.
5. Compare the two numbers and conclude the final answer.

This example demonstrates TableRAG’s capability in understanding, reasoning, and planning, which goes beyond simple table manipulation and Text2SQL.

---

### Decision · Program_Chairs · 2024-09-25

**Decision:**

Accept (poster)

**Comment:**

LM-based table retrieval suffers from scalability challenges in case of large schema and tables.  This submission introduces TableRAG reduces these scalability challenges by query expansion and retrieval mechanisms.  The submission and rebuttal include a thorough  evaluation on benchmarks sourced from real-world datasets and synthetic data from TabFact.  TableRAG shows better performance and lower LM token costs.

This submission represents a nice systems-level work that seems effective and intuitive. Because of the nature of systems built around LMs, the paper does not fit the template of algorithmic ML papers with high creativity or intellectual depth.  In aggregate, reviewers found the paper borderline.  The authors reported on substantial additional experiments during rebuttal, and resolved several issues reported by reviewers.

The authors must publish code and data if the paper is accepted.